# Macronutrient Intake, Sleep Quality, Anxiety, Adherence to a Mediterranean Diet and Emotional Eating among Female Health Science Undergraduate Students

**DOI:** 10.3390/nu15132882

**Published:** 2023-06-26

**Authors:** Germán Díaz, Sonsoles Hernández, Almudena Crespo, Alina Renghea, Hugo Yébenes, María Teresa Iglesias-López

**Affiliations:** Health Sciences Faculty, Universidad Francisco de Vitoria, Carretera Pozuelo-Majadahonda km 1800, Pozuelo de Alarcón, 28223 Madrid, Spain; german.diaz@ufv.es (G.D.); s.hernandez@ufv.es (S.H.); a.crespo.prof@ufv.es (A.C.); a.renguea@ufv.es (A.R.); hugo.yebenes@ufv.es (H.Y.)

**Keywords:** female undergraduate students, COVID, emotional eating, sleep quality, anxiety

## Abstract

Introduction: COVID-19 provoked a myriad of challenges for people’s health, poor life satisfaction and an unhealthy diet that could be associated with serious negative health outcomes and behaviours. University is a stressful environment that is associated with unhealthy changes in the eating behaviours of students. The association between diet and mental health is complex and bidirectional, depending on the motivation to eat; emotional eaters regulate their emotions through the increased consumption of comfort foods. Objective: The aim of this study was to compare the nutritional habits, alcohol consumption, anxiety and sleep quality of female health science college students. Material and methods: A cross-sectional study of 191 female undergraduate students in Madrid was used. Their body mass index and waist hip ratio were measured. The questionnaires used included the Mediterranean Diet Adherence test, AUDIT, Emotional Eater Questionnaire, Pittsburgh Sleep Quality Index, and Food Addiction, Perceived Stress Scale and STAI questionnaires. Results: We observed a high intake of protein, fat, saturated fatty acids and cholesterol. Overall, 9.5% never had breakfast, and 66.5% consciously reduced their food intake. According to Pittsburgh Sleep Quality Index, they mainly slept 6–7 h, and 82% presented with a poor sleep quality; 13.5% presented moderate–high food addiction, and 35% had moderate Mediterranean Diet Adherence score. Conclusion: Female students’ macronutrient imbalances were noted, with a high-level protein and fat intake diet and a low proportion of carbohydrates and fibre. A high proportion of them need alcohol education and, depending on the social context, they mainly drank beer and spirits.

## 1. Introduction

Young adulthood is an important period to establishing good habits and healthy eating patterns to prevent the risk of chronic diseases that will affect people’s quality of life [1]. University students seem to be influenced by some individual factors and their social networks [2]. Socio-economic statuses influence diet quality; a good diet can reduce adolescent obesity and non-communicable diseases in the future. University is a stressful environment that is associated with unhealthy changes in eating behaviours in students due to university characteristics, such as living arrangements or academic schedules, which influence the relationships between individuals and their eating behaviours [2,3].

Protein requirements depend on age, fat free mass, physical activity and the severity of the disease. The recommended daily amount of protein that must be consumed is 0.8 g/kg/day in healthy individuals [4,5]. High-protein diets are characterised by a protein content that is above the recommended values [6]. With respect to protein intake, no negative effects on kidney function have not been found after the consumption of 2.51–3.32 g/kg/d for one year, which is probably due to their high fibre intake of more than 30 g fibre per day [7]. Related to that, Knight et al. observed that among females, high-protein intake was not associated with renal function decline in those with normal renal function; nevertheless, it may accelerate renal function decline in females with mild renal insufficiency with non-dairy intake [8]. Jhee et al. on the contrary, noted that a high-protein diet has a deleterious effect on renal function in the general population [9]. A high-protein intake is considered to increase the risk of renal hyperfiltration in individuals by inducing glomerular hypertrophy and an increased risk of hyperfiltration [10]. Some food group characteristics of a proinflammatory diet, such as meat or processed foods, can produce health problems and increase the risk of some non-communicable diseases (obesity, hypercholesterolemia or diabetes) [11].

COVID-19 provoked a myriad of challenges for people’s health, causing them to have a low level of life satisfaction and an unhealthy diet that could be associated with serious negative health outcomes and behaviours [12,13]. Depending on their socio-economic status, some individual’s habits are affected in different ways. The combination of stress, anxiety and depression due to this pandemic situation also had an impact on eating behaviours [14].

The Perceived Stress Scale-14 (PSS-14) was designed to assess the degree of stress people felt in unpredictable and out-of-control situations. It has fourteen items, with seven negative and seven positive items [15].

Emotional distress and anxiety appear to play a major role in the choice, quality and quantity of food intake, which could lead to an increase in BMI and lead to the consumption of unhealthy foods [16,17]. The association between diet and mental health is complex and bidirectional [18]. As reported by Macht [16], depending on the motivation to eat, emotional eaters regulate their emotions via the increased consumption of comfort foods. A study of adult females reported that those with high-level anxiety showed more unhealthy eating and food cravings [17]. Lovan et al. observed that female students appeared to have lower susceptibility to their internal bodily signal signals of hunger and satiety and a higher reliance on emotions to initiate or end eating [19]. More emotional changes occur in females during this period, and also, the researchers observed an association between alcohol consumption and the perception of apathy and anxiety related to food cravings between meals [20]. Scott et al. noted that, in stressful situations and anxiety, people tend to regulate their emotions via eating [21]. Gonçalves et al. observed in their study that only females reported having a food addiction [22].

The Pittsburgh Sleep Quality Index (PSQI) is an instrument used to assess subjective sleep quality and is also a reliable and valid instrument for university students [23].

Due to globalisation, the importation of Western habits and shifts in lifestyle are probably among the potential drivers away from the traditional Mediterranean diet (MD). There are different indices used to measure Mediterranean Diet Adherence (ADM), and the indices are generally the food groups traditionally consumed by Mediterranean populations. However, other foods from non-Mediterranean areas and locally consumed ones may be inappropriately computed with these indices [24]. Regarding alcohol consumption, for some young people, both food and alcohol are a source of pleasure of their social lives [21].

The aim of this study was to compare the nutritional habits, alcohol consumption, anxiety and sleep quality among female health science university students.

## 2. Material and Methods

### 2.1. Participants

This cross-sectional study was conducted in February 2022 with health science undergraduate students from Madrid. Participation was voluntary, and anonymity was ensured. The mean age of students was 19.8 ± 1.5 years. Individuals were excluded if they did not properly complete the questionnaires. Informed consent was provided, and the test was completed online, with data only collected during college hours, and any incidents and all the results were anonymously analysed. The ‘snowball’ sampling method was used to recruit more participants. The inclusion criteria were that they must be nursing students (age < 24 years). We discarded males because females were more representative of the nursing degree at the university of precedence. Finally, the total sample studied was 191 female undergraduate students (Figure 1).

Prior to the start of the study, participants were informed of the purpose of the research, and the study protocol and design were approved by the Ethics Committee of the University Francisco de Vitoria (UFV 2022-26) and fully complied with the 1964 Helsinki Declaration and its later amendments.

### 2.2. Anthropometrics Measurements

Anthropometric measurements (Table 1) were carried out using calibrated digital scales, SECA^®^ 840 and 877 (SECA Vogel & Halke, Hamburg, Germany), as well as portable stadiometers, SECA^®^ 214 and 217 (SECA Vogel & Halke, Hamburg, Germany). Students’ weights were measured to the nearest 100 g unit (0.1 kg) in kilograms, barefoot and wearing light clothes, and their statures were measured to the nearest millimetre (0.1 cm), with the subject standing fully erect with their feet together, head in the Frankfort plane and arms hanging freely. Body mass indexes (BMI) were calculated using the formula weight (kg)/[height (m)^2^]. Subjects were classified as underweight (BMI < 18.5 kg/m^2^), had a normal weight (BMI 18.5–24.9 kg/m^2^) or overweight/obese (BMI > 25 kg/m^2^) based on the World Health Organization (WHO) criteria [25,26]. Their waist circumferences were measured using an anthropometric measuring tape at a horizontal plane midway between the lowest rib and the iliac crest. A cut-off point for risk identification for metabolic diseases of a waist-to-hip ratio > 0.85 for females [27] was used.

Table 1 shows the anthropometric measures of the students, and we observed that they mainly had a normal weight (74.5%), with 14% being underweight and 11.5% being overweight/obese.

### 2.3. Dietary Assessment

Food intake was assessed using three-day food records (two weekdays and one weekend day/holiday) after receiving instructions during class time. The dietary intake in terms of a selection of macronutrients and the Healthy Eating Index were determined using Spanish DIAL^®^ software (Alce Ingeniería, Las Rozas de Madrid, Spain) [28].

### 2.4. Questionnaires

(a) The Mediterranean Diet Adherence (ADM), which was originally developed for the Spanish population [29], was used. This questionnaire contains 14 items representing the dietary components of the MD. Each item was scored as zero (no adherence to a dietary component) or one (maximal adherence to a dietary component), and the range of the index was 0–14. The cut-offs to classify subjects were high (>9), moderate (7–9) or low (0–7) according to the ADM.

(b) For alcohol consumption, we used the limits established by García-Carretero [30] for the interpretation of the AUDIT score among Spanish university students. The cut-off for females was six, and for high-risk consumption, a score of thirteen for both males and females indicated a probable alcohol dependence syndrome.

(c) The State-Trait Anxiety Inventory (STAI) adapted by Guillén-Riquelme and Buela-Casal [31] is composed of 40 items that are used to evaluate two different concepts: anxiety as a state (a transitory emotional response) and anxiety as a trait (a constant anxious condition). Twenty items of each subsection use a Likert scale of four categories from zero (almost never) to three (very often/almost always).

(d) The Pittsburgh Sleep Quality Index (PSQI) is a 19-item questionnaire based on seven components of sleep (quality, onset latency, duration, efficiency, disturbance, use of sleep medication and daytime dysfunction). Each component scores from zero to three, with higher scores indicating poorer sleep. Seven component scores are summed for a global score ranging from zero to twenty-one, and a score above five indicates poor sleep quality [32].

(e) The Perceived Stress Scale Spanish version (PSS-14) is a 14-item questionnaire with a five-point response scale, where 0 = never, 1 = almost never, 2 = once in a while, 3 = often and 4 = very often. The total score is obtained by reversing the scores of items 4, 5, 6, 7, 9, 10 and 13 (0 = 4, 1 = 3, 2 = 2, 3 = 1 and 4 = 0, respectively), which are positively stated items. A higher score indicates a higher level of perceived stress, and the range of the scores is 0–56 [33].

### 2.5. Statistical Analysis

To verify normal distribution of the data, a Kolmogorov–Smirnov test was conducted. Due to the distribution, non-parametric analysis was performed. Twenty-fifth and seventy-fifth median percentiles were calculated for descriptive analysis.

A one-sample t-test and Cohen’s d were calculated to compare the mean value and recommended daily intake (RDI). The Kruskal–Wallis test was performed to analyse differences among groups. If the Kruskal–Wallis test had a *p*-value > 0.05, a Mann–Whitney test was calculated to compare paired groups.

Spearman’s rho correlation was used to compare the association between variables.

All statistical analyses were performed using SPSS 29.0 (IBM Corporation, Endicott, NY, USA), and the level of statistical significance was set at *p* < 0.05.

## 3. Results

In Table 2, we show the macronutrient intake data. When we compared the energy intake with the BMI of students, the median values did not show differences between underweight and overweight students (1610 kcal). Only slight, but not significant, differences were observed between the twenty-fifth and seventy-fifth percentiles. In this case, the underweight students had greater energy intake than the overweight students did. According to protein %E, the median was 17.8%, which is a higher value than that of the recommendations. Only 9.4% of students had a protein intake in line with the recommendations. With respect to fat %E, the median was 37.7%, which is a value higher than that which is recommended. With respect to this fat intake, 66.5% of female students had a fat %E higher than the recommended value. Regarding saturated fatty acids (SFA) %E and cholesterol, both values were also higher than those of the recommendations, as shown in Table 2. On the other hand, the values of carbohydrates (CH) %E did not reach those that are recommended, being only 1%. Fibre intake was also scarce, with only 12% of students reaching the recommended fibre intake values. The effect size was high for carbohydrate %E and fibre intake, medium for protein, fat and SFA %E, and low for cholesterol.

When we compared the macronutrient intake data according to BMI (Table 3), significant differences in fat %E (U = 1262, *p* = 0.03) and cholesterol (U = 12301, *p* = 0.01) were observed between the underweight and overweight/obese respondents.

In Table 4, we show that the students mainly breakfasted every day, but 8.9% skipped breakfast. No differences were observed with respect to the reduction or not of food intake, and they mainly drank beer and spirits.

Table 5 shows that according to the results of PSQI, 82% of the female students had a poor sleep quality. With respect sleep efficiency, duration and subjective sleep quality, 16.5% had scores of less than 75%, and half of the students slept between 6–7 h/day, 26% slept ≤ hours/day, and finally, half of the students had quite good sleep quality, but among the 26%, it was bad or very bad. Finally, we highlight that 25% of female nursing students took sleep medication.

Table 6 shows that 13.5% presented with moderate–high food addiction, 56% had a medium level of adherence to the ADM, and 80.1% of participants need alcohol education.

In Table 7, we show that participants who consumed fatty acids (FA) had higher levels of energy and fat intake due to the correlation between them, but less perceived stress, which was correlated inversely. Individuals with high levels of perceived stress consumed less energy and protein; on the other hand, a positive correlation between stress and anxiety states was observed. Finally, the anxiety trait was correlated with FA and fat (fat %E and SFA %E), and participants who consumed FA were less likely to adhere to the ADM.

In Table 8, we show that FA was inversely correlated with ADM, which is in concordance with a significant BMI–fat %E relationship. An inverse correlation has also been observed between BMI and fibre; in this case, the higher the BMI is, the lower the level of fibre intake of the students is. Finally, participants with better ADM scores had higher-quality sleep, according to the correlation observed between the PSQI and ADM.

## 4. Discussion

Changes in eating patterns are occurring worldwide. The purpose of this cross-sectional study was to identify the nutritional habits, alcohol consumption, anxiety and emotional eating and sleep quality of university health science students.

The present study highlights the elevated proportion of female students who do not meet the dietary guidelines, but differences in methodology and age classification occasionally make comparisons problematic. It is not the first time that poor eating habits among university students [34] have been described. In this study, the macronutrients %E was markedly unbalanced as compared to the updated reference values of the EFSA and ANIBES studies [35,36]; protein %E (17.8%), fat %E (37.7%) and SFA %E (12.1%) were significantly higher than the recommended values, and it was lower for carbohydrates %E (37.7%). The percentages calculated serve to classify this dietary pattern within the so-called low-carbohydrate–high-protein diet spectrum (LCHP diet) [37].

These results were consistent with previous studies carried out among other university students [38,39]. With respect to protein %E, only 9.4% were within the recommended range, which was also observed in the ANIBES study [40]. However, if we compared our results with those recommended by the Institute of Medicine (10–35%), the values were under the lower limit. The Institute of Medicine, in fact, reported very a high level of protein intake (>35% E) without negative effects [6]. On the other hand, it is accepted that excessive long-term protein consumption may counteract an adequate energy profile [40]. Protein-rich diets have been demonstrated to be effective at lowering a person’s body weight, without negatively affecting cardiovascular health markers, such as the cholesterol and triglyceride levels in plasma. However, there are some concerns about their potential long-term negative effects, as studies carried out using animal models show kidney, hepatic and cardiovascular alterations [41,42], while, in humans, the evidence remains inconclusive [43].

Our data show a significant inverse correlation between BMI and protein %E, as is expected due to the metabolic effects of a protein-rich diet. It has been shown that high-protein diets promote body weight reduction over the long-term, which is in part due to an increase in the sleeping metabolic rate (SMR) [44,45], and also, a significant increase in dietary energy expenditure (DEE) [46,47] in comparison with those of normal composition diets. Additionally, a long-term high-protein diet (%E P > 20%) in healthy individuals promotes more satiety in comparison with that of diets with a standard protein composition (%E P < 15%) [37] due to the release of GLP-1 and insulin [48].

In this study, female students’ macronutrient imbalances were noted with high-protein intake diets, which were inversely correlated with BMI. In a study with a Chinese population, high-protein meals led to reductions in the total energy intake and were associated with a lower body weight, BMI, waist circumference, higher adiposity and correlated with saturated fatty acids [49]. When comparing macronutrient intake with BMI, the results were similar to Chinese–American university students [39]. No differences were detected between the underweight and overweight students, but we observed a greater energy intake in the underweight students than we did in the overweight students.

We also found a negative trend in fat intake, with a fat %E of 37.7%, which is a value that is above the recommendations [50]. With respect to fat intake, 66.5% of female students had a fat %E that was higher than the recommended value. This is in concordance with the fat intake level in Spain, which is at the upper end of the recommended value, and certain types of dietary fat may contribute to cardiovascular diseases [40]. Regarding SFA %E and cholesterol, both values were at the upper end of the recommendations. Even so, the EFSA, FAO and the Spanish Federation of Food, Nutrition and Dietetic Societies (FESNAD) have recommended a maximum intake of 10% E for SFA in the adult population [51,52].

The Spanish population, however, is well below the lower limit of CH, which is considered to be a poor indicator of their present diet quality [40]. In this study, CH %E did not reach the value recommended by the EFSA of 45–60%E, whereas the SENC recommended 50–60%E. However, as observed in the ANIBES study, the Spanish population was below the lower limit, and that is an indicator of poor diet quality [40,53,54]. Our values for CH %E did not reach the recommendations, and similar results were observed among Spanish females. Their fibre intake level was also poor, with only 12% of students reaching the recommended fibre intake values. When we compared the above intakes according to BMI, significant differences were observed between the students with a normal weight and those with overweight/obesity. According to their BMI, the relationship between underweight and overweight/obese students was significant only for fat %E intake (*p* = 0.007).

Eating breakfast may be an important dietary behaviour for cardiometabolic health. Therefore, consuming breakfast is common among young adults and is the most frequently skipped meal [55]. Ferrara et al., during COVID pandemic period, observed that, in 2021, 42.9% more females ate breakfast than they did in the beginning and pre-pandemic periods [56]. In this study, this daily unhealthy habit was 8.9%, a higher value than that which was observed in a Basque university, which was 5% [57]. There is strong evidence of an association between breakfast skipping and overweight/obesity [48], but in this study, there was a lack of association between overweight/obesity and skipping breakfast. In a cross-sectional study, students who skipped breakfast were more vulnerable to alcohol consumption and its toxic effects [58]. Furthermore, in this study, a lack of a significant association was observed between both unhealthy habits.

Regarding drinking behaviour, 80.3% were alcohol drinkers, who mainly drank beer (30.9%) and spirits (20.9%), which is similar to the results observed by Scott et al. among young adults in the UK [21]. In this study, underweight and overweight/obese students drank more alcohol than the others did, and significant differences (*p* < 0.05) were observed between the underweight and normal weight students according to type of beverage (beer and spirits) consumed. More than 2/3 of respondents in this study need alcohol education, and depending on their social context, they responded that they drank mainly beer and spirits with friends. As reported by Scott et al. [21], on their days off, young adults visiting take away food shops consume large amounts of alcohol and often also eat unhealthy food.

Low-quality sleep seems to be particularly common among university students [59,60], with reported prevalence rates of 50–70% for low-quality sleep being assessed with the PSQI [32]. Descriptive statistics in this study showed that up to 82% of the sample reported low-quality sleep according to the PSQI, and 26% had a habitual sleep duration < 7 h, which is the minimum amount of sleep conventionally recommended for young adults and adults [61]. Finally, we highlight that 25% of female nursing students took sleep medication. In this study, participants with better ADM scores had significantly high-sleep quality.

Another finding of this study was that 56% of female students presented mid-range AMD scores, similar as previously observed in Spanish nursing students; the health benefits of a Mediterranean diet across the Mediterranean basin have been demonstrated, and it is important to promote the AMD among future health professionals [34,62,63].

It has been observed that stress and emotional conditions modulate eating behaviour [34]. This is in concordance with the significant relationship detected between BMI and fat %E and lower intake of fibre when the BMI increased. Dakanalis et al. also found an association between BMI and eating behaviours [64], and some negative dietary habits could be due to the stressful situation during the pandemic situation [20]. ElBarazi and Tikamdas found a positive association between fatty food intake, stress, anxiety and BMI, which is contrary to the healthy and adequate nutrient intake that benefits mental health [65]. Among Portuguese university students, no differences were obsrved between food addiction and BMI during the pandemic situation, which could have promoted healthier or unhealthier eating habits [22].

The perceived stress score among female students was 23, which indicated a medium level of stress, where the stress ranged from 0 to 56 [33]. In this study, perceived stress was inversely correlated with the intake of energy and protein %E intake. Similarly, Pal et al. suggested that stress negatively affects emotional regulation and may promote unhealthy eating behaviour and hedonic eating [66].

Some limitations should be considered. One of them is the preponderance of female participants, which makes generalisation for both sexes difficult. We avoided male university students due to the small size of the accessible population. The study comprises nursing students, and during the period of study, the students were experiencing stressful situations because it was conduced the examination period.

## 5. Conclusions

In this study, female students’ macronutrient imbalances were noted, and they consumed a high-protein and fat intake diet and a low proportion of carbohydrates and fibre. A high proportion of females need alcohol education, and depending on the social context, they mainly drank beer and spirits.

It is necessary to promote healthier eating among university students and to address food addiction in this group. After the pandemic, it has become necessary to increase the promotion of healthy versus unhealthy eating habits.

Future studies are essential to assessing the underlying mechanisms linking skipping breakfast with the incidence of frequent alcohol intake, food addiction, stress and anxiety among university students.

## Figures and Tables

**Figure 1 nutrients-15-02882-f001:**
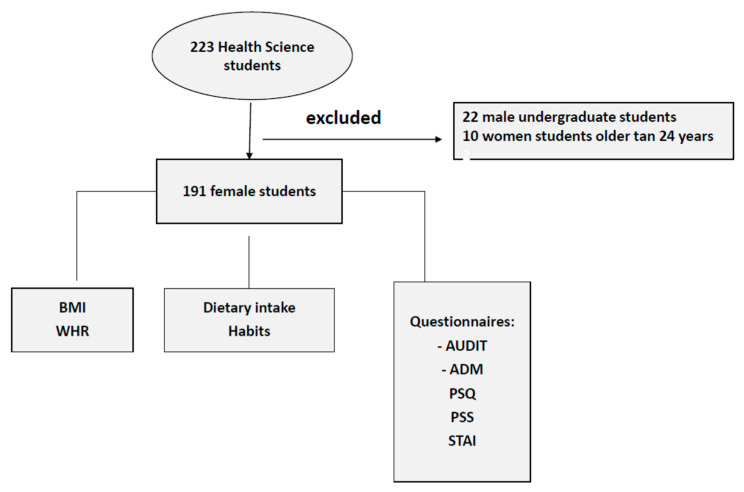
Flowchart of the study.

**Table 1 nutrients-15-02882-t001:** Anthropometric measurements of participants in the study.

		N	%
(kg/m^2^)	<18.5	28	14.0
	18.5–24.9	149	74.5
	>25	23	11.5
	M *	20.4	
	P25	19.2	
	P75	22.7	
WHR *	<0.85	185	92.5
	>0.85	15	7.5

* M = median; WHR = waist/hip ratio.

**Table 2 nutrients-15-02882-t002:** Energy and macronutrient energy intake in nursing students.

	E (kcal)	%E P	%E CH	Fibre (g)	%E F	%E SFA	Cholesterol (mg)	HEI *
M	1692	19.5	39.4	17.9	41.6	13.7	311.4	59.4
P25	1456.5	17.7	31.8	14.2	35.5	11.3	230	41
P75	2158.5	23.4	43.6	23.4	46.3	17.6	356.5	42.3
RDI *		15	50	30	35	<10	300	
*p*		<0.001	<0.001	<0.001	0.003	<0.001	0.219	
ES		0.763	1.609	1.213	0.205	0.31	0.056	

M: median; E: energy; %E P, CH, F and SFA, which are % energy of proteins, carbohydrates, fat and saturate fatty acids, respectively; * HEI: Healthy Eating Index; RDI: recommended daily intake; *p*: *p*-value; ES: effect size.

**Table 3 nutrients-15-02882-t003:** Differences between %E of macronutrients, fibre and cholesterol intake according to Body Mass Index (BMI).

		BMI				
		UW	NW	OW-O	*p*	Kruskal–Wallis
		(%) *	(%) *	(%) *		
%E P	≤15%	4	10.5	8.7	0.61	
	≥15%	96	89.5	91.3		0.980
%E CH	≤50%	96	99.3	100	0.26	
	≥50%	4	0.7	0		2.681
fibre	<30 g	84	87.4	95.7	0.38	
	≥30 g	16	12.6	4.3		1.927
%E F	≤35	44	35.7	8.7	0.01	
	>35	56	64.3	91.3		8.864
%E SFA	≤35%	12	16.1	21.7	0.69	0.733
	≥35%	88	83.9	78.3		
cholesterol	≤300 mg	68	42	65.2	0.02	77,872
	≥300 mg	32	58	34.8		

* UW = underweight; NW = normal weight; OW-O = overweight/obesity.

**Table 4 nutrients-15-02882-t004:** Some habits of nursing students.

		N	%
Breakfast	Yes	131	68.1
	No	17	8.9
	Sometimes	43	23
I reduce the amount of food intake	Yes	133	66.5
	No	75	68
Alcohol intake	no drink	37	18.5
	wine	11	5.5
	beer	72	36
	spirits	80	40

**Table 5 nutrients-15-02882-t005:** Results from the Pittsburgh Sleep Quality Index (PSQ).

PSQI	N	%
Good sleep quality	36	18
Bad sleep quality	164	82
Sleep medication		
Never last month	150	75
Less than once/week	28	14
1–2 times/week	14	7
≥3 times/week	8	4
Sleep efficiency		
>85%	117	58.5
75–84%	50	25
65–74%	22	11
<65%	11	5.5
Sleep duration (hours)		
>7	43	21.5
6–7	105	52.5
5–6	43	21.5
<5	9	4.5
Subjective sleep quality		
Very good	43	21.5
Quite good	105	52.5
Quite bad	45	22.5
Very bad	7	3.5

**Table 6 nutrients-15-02882-t006:** Descriptive data of questionnaires: Food Addiction (FA), Adherence to Mediterranean diet (ADM) and AUDIT.

		N	%
FA	no addiction	139	69.5
	low addiction	34	17
	moderate addiction	17	8.5
	high addiction	10	5
ADM	low adherence	78	39
	medium adherence	112	56
	high adherence	10	5
AUDIT	alcohol education	153	80.1
	alcohol education + monitorisation	38	19.8

ADM: Mediterranean Diet Adherence.

**Table 7 nutrients-15-02882-t007:** Spearman correlations between variables in the study.

	1	2	3	4	5	6	7	8	9	10	11
Energy	-										
P	0.651 **	-									
CH	0.682 **	0.254 **	-								
fibre	0.200 **	0.225	0.113	-							
fat	0.735 **	0.458 **	0.528	−0.070	-						
AGS	0.592 **	0.541 **	0.364	−0.012	0.685 **	-					
cholesterol	0.408 **	0.484 **	0.172	0.010	0.333 **	0.489 **	-				
PSS-14	−0.149 *	−0.166 *	−0.093	0.018	−0.128	−0.078	−0.088	-			
FA	0.158 *	0.061	−0.002	0.017	0.167 *	0.136	0.137	0.105	-		
Anxiety state	−0.106	−0.141	0.032	−0.084	−0.105	0.004	−0.027	0.245 **	−0.035	-	
Anxiety trait	0.079	−0.015	0.053	0.156 *	0.190 **	0.162 *	0.037	−0.106	0.216 **	0.097	-

P: protein; CH: carbohydrate; SFA: saturated fatty acid; PSS-14: Perceived Stress Scale-14 items; FA: food addiction; PSQI: Pittsburgh Sleep Quality Index. * *p* < 0.05; ** *p* < 0.01.

**Table 8 nutrients-15-02882-t008:** Spearman correlations between fat, fibre, ADM and PSQI.

	1	2	3	4	5	6
Fat %E	-					
SFA %E	0.411 **	-				
BMI	0.168 *	−0.058	-			
fibre	−0.372 **	−0.183 *	−0.179 *	-		
ADM	−0.068	0.028	0.088	−0.037	-	
PSQI	0.080	0.050	−0.004	−0.064	0.200 **	-

FA: food addiction; SFA %E: % energy of saturated fatty acids; BMI: body mass index; ADM: Mediterranean Diet Adherence; PSQI: Pittsburgh Sleep Quality Index. * *p* < 0.05; ** *p* < 0.01.

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
