# Peer review of "Macronutrient Intake, Sleep Quality, Anxiety, Adherence to a Mediterranean Diet and Emotional Eating among Female Health Science Undergraduate Students"

_nutrients, 2023, doi:10.3390/nu15132882_

Round 1
Reviewer 1 Report
Thank you for providing me with the opportunity to review this manuscript. The topic presented in the paper is interesting, as is Mediterranean Diet and Emotional Eating among Health Science Undergraduate Female Students during the Second Wave of COVID-19
The hypotheses tested overall propose an appealing analysis.
I would like to address and solicit further discussion on some aspects that I consider essential. I hope that by sharing my views on this study, the authors can improve the quality of their publication. My recommendations will be presented in the order of the paper.
The technical description is timely and accurate. However, some points, if attended to by the authors with some additions, could further improve the quality of the work.
1)Rewrite and subdivide the abstract according to journal standards
2) The introduction should be revised in translation, with a more synthesis, some passages are not smooth and there is too much redundancy and repetition.
Line 42 -Reduce the space between 'than 30 g'.
3) The description of the instruments is not accurate also needs, in my opinion, to be expanded.
4) the discussion should be revised in traslation
5) The graphic and tabular layout is not well enough cared for. They should be fixed. The picture is unclear. The picture should be clear and have a high resolution
5)Conclusions should be expanded. Study limitations are missing
We recommend consulting and adding the following articles:
https://doi.org/10.3390/ijerph19095550
https://doi.org/10.3389/fpsyg.2022.941784
https://doi.org/10.3390/nu14193989
https://doi.org/10.1515/med-2021-0291
As far as language is concerned. Grammar revision by a native English speaker is required
Author Response
Dear reviewers, first, we want to thank you for all the contributions to our manuscript. We are sure that all your suggestions contribute to improve the quality of our study as well as make the dissemination of the results more understandable for the audience interested. Below we detail point by point the changes made to the manuscript after your reviews. All changes in this new version of the manuscript are marked. We apologize for the mistake. We have carefully checked the document and we have improved the writing in the final version of the manuscript.
1) Rewrite and subdivide the abstract according to journal standards
According to the Reviewer consideration, we have subdivided the abstract as it has been suggested in the new English version of the manuscript.
2) The introduction should be revised in translation, with a more synthesis, some passages are not smooth and there is too much redundancy and repetition.
According to the Reviewer suggestion, we have sent the paper to a new translation with MDPI translators. Also, according to Reviewer recommendations, we corrected repetition in the Introduction.
Line 42 -Reduce the space between 'than 30 g'.
We appreciate the suggestion of the Reviewer, and we reduced that space.
3) The description of the instruments is not accurate also needs, in my opinion, to be expanded.
We have modified the description of alcohol consumption as Reviewer suggested.
4) the discussion should be revised in translation
According to the Reviewer suggestion, we have sent the paper to a new translation with MDPI translators.
5) The graphic and tabular layout is not well enough cared for. They should be fixed. The picture is unclear. The picture should be clear and have a high resolution.
Following the Reviewer recommendation, we have improved the picture.
5)Conclusions should be expanded. Study limitations are missing.
We appreciate the Reviewer comment and according to their suggestion we expanded the conclusions.
However, in lines 417-421 we previously comment the study limitations.
We recommend consulting and adding the following articles:
Following the Reviewer suggestion we consulted that articles, to improve the quality of the study.
https://doi.org/10.3390/ijerph19095550
https://doi.org/10.3389/fpsyg.2022.941784
https://doi.org/10.3390/nu14193989
https://doi.org/10.1515/med-2021-0291
Reviewer 2 Report
Macronutrient Intake, Sleep Quality, Anxiety, Adherence to a 2 Mediterranean Diet and Emotional Eating among Health 3 Science Undergraduate Female Students during the Second 4 Wave of COVID-19
Thank you for the chance to review the above-mentioned paper.
Some comments:
I wonder if the title is misleading as there is little to no mention of the impact of Covid-19 on the variables collected.
It is worth checking the English expression in this paper as it is quite difficult to read eg in the introduction:
“It has been observed with respect to protein intake, no negative effects on kidney function have not been found after consumption of 2.51–3.32 g/kg/d during one year, 41 probably due to their high fibre intake, more than 30 g fibre per day [7].”
It would be clearer to write: No negative effects on kidney function have been observed after the consumption of 2.51–3.32 g/kg/d of protein over the course of one year. This lack of effect could be due to a concurrent high fibre intake, i.e. more than 30 g of fibre per day [7].
On lines 51-52, you have the following sentence López- 51 Moreno et al. observed in nursing students that a high intake of proteins was related to 52 greater academic performance [12]. This is immediately after all the negatives. However, before this sentence would make more sense.
However, there are many other examples and the whole document should be reviewed.
In the introduction, you mention several of the questionnaires that you used but without context, you might as well take them out of the introduction and put them in the method section.
Methods
What does “with any incidents” line 96 mean?
Line 101 add the word “nursing” between female and undergraduate students.
Am I reading figure one correctly that you were able to get every nursing (health science) students that wasn’t male or over 24 to complete the questionnaire and be weighed? This is extraordinary! Unless they were all required to partake in their study as part of their course?? If this is so you should state this. If there was potential for more, you should put these in.
Results:
The first time you mention %E you should define it I think this is on line 175
In the introduction you discuss protein as 0.8g/kg/d not as a percentage but in the results section it is expressed as a percentage. You then say that the protein intake higher or in line with recommendations, but this is difficult for the reader to confirm given the different units used. It is expressed in the tables but it should be spelt out in the introduction. What percentage are you using as the recommendations?
Line 204 remove the words “we highlight that”... Start the sentence as 25% of female….
Sometimes the table # is bolded and sometimes it is not bolded. In the text section of the results unbold the title and its number.
Table 7 should have the numbers listed next to the variables to make the table easier to read. There is no SFA variable in Table 7 although it is mentioned underneath. In the text written 214-219 you need to add Table 8 when you start talking about its results.
In line 214 you say “In Table 7 we show that participants with fatty acids (FA) had higher energy and fat..” FA stands for food addiction. There was no correlation with PSS-14 the measure of stress.
Under table 8 you mention the variable food addiction although it is not in this table only put in the abbreviations of the variables actually mentioned in the table.
Discussion:
Chine-American I don’t think this is correct do you mean Chinese-American (line266). Explanation around why you think you found a lower intake in overweight individuals than the underweight individual would be helpful after the sentence finishing on line 268.
Please review the English expression throughout the discussion it is confusing to read.
The quality of the english lanugage needs to be improved in both the introduction and the discussion.
Author Response
Dear Reviewer, first, we want to thank you for all your contributions to our manuscript.
We apologize for the mistake and we have deleted in the title “Covid-19”
It is worth checking the English expression in this paper as it is quite difficult to read eg in the introduction:
“It has been observed with respect to protein intake, no negative effects on kidney function have not been found after consumption of 2.51–3.32 g/kg/d during one year, 41 probably due to their high fibre intake, more than 30 g fibre per day [7].”
It would be clearer to write: No negative effects on kidney function have been observed after the consumption of 2.51–3.32 g/kg/d of protein over the course of one year. This lack of effect could be due to a concurrent high fibre intake, i.e. more than 30 g of fibre per day [7].
We appreciate the suggestion of the Reviewer and we rewritte that phrase.
On lines 51-52, you have the following sentence López- 51 Moreno et al. observed in nursing students that a high intake of proteins was related to 52 greater academic performance [12]. This is immediately after all the negatives. However, before this sentence would make more sense.
Following the Reviewer recommendation, we have deleted that paragraph.
However, there are many other examples and the whole document should be reviewed.
We apologize for the mistakes; we presented the new English version properly revised by MDPI translator.
In the introduction, you mention several of the questionnaires that you used but without context, you might as well take them out of the introduction and put them in the method section.
Methods
What does “with any incidents” line 96 mean?
We understand the appreciation of the Reviewer, and we deleted the entire phrase in order to clarify the understanding.
Line 101 add the word “nursing” between female and undergraduate students.
Based on the Reviewers comment, we have carefully reviewed in line 101 your suggestion, but we added “health science” instead of “nursing” because the female were from health sciences.
Am I reading figure one correctly that you were able to get every nursing (health science) students that wasn’t male or over 24 to complete the questionnaire and be weighed? This is extraordinary! Unless they were all required to partake in their study as part of their course?? If this is so you should state this. If there was potential for more, you should put these in.
We agree with the Reviewer consideration about the number of students included. Certainly, 223 Health students were enrolled in the study, but finally only 191 females were included, according to the inclusion criteria “no older than 24 years” and we excluded male due to only 22 males accepted to participate in the study. We didn’t consider that population because of the big difference between sexes.
Results:
The first time you mention %E you should define it I think this is on line 175
Regarding the comment related with %E and according to Reviewer suggestion we included percentage of protein energy (%E)
In the introduction you discuss protein as 0.8g/kg/d not as a percentage but in the results section it is expressed as a percentage. You then say that the protein intake higher or in line with recommendations, but this is difficult for the reader to confirm given the different units used. It is expressed in the tables but it should be spelt out in the introduction. What percentage are you using as the recommendations?
With respect to the Reviewer's personal comment, the information required is reflected in the Introduction as 12-15% of protein energy according to recommendations.
We apologize for the difficult to understand protein recommendations, we used as %E protein recommendation 12-15%.
Line 204 remove the words “we highlight that”... Start the sentence as 25% of female….
We agree with the Reviewer consideration and we start the phrase according to their consideration.
Sometimes the table # is bolded and sometimes it is not bolded. In the text section of the results unbold the title and its number.
We apologize for the mistake; we have always put table unbolded.
Table 7 should have the numbers listed next to the variables to make the table easier to read. There is no SFA variable in Table 7 although it is mentioned underneath. In the text written 214-219 you need to add Table 8 when you start talking about its results.
We apologize for the mistake, we deleted AGS and we added SFA
In line 214 you say “In Table 7 we show that participants with fatty acids (FA) had higher energy and fat..” FA stands for food addiction. There was no correlation with PSS-14 the measure of stress.
We apologize for the mistake; we deleted fatty acids and write food addiction (FA). And regarding Reviewer consideration there was no correlation with PSS-14 the measure of stress.
Under table 8 you mention the variable food addiction although it is not in this table only put in the abbreviations of the variables actually mentioned in the table.
We apologize for the mistake; we deleted the comments of food addiction.
Discussion:
Chine-American I don’t think this is correct do you mean Chinese-American (line266). Explanation around why you think you found a lower intake in overweight individuals than the underweight individual would be helpful after the sentence finishing on line 268.
We apologize for the mistake, in the new English version is properly written.
Please review the English expression throughout the discussion it is confusing to read.
We apologize for that and according to Reviewer recommendations, we revised English expression
Comments on the Quality of English Language
The quality of the english lanugage needs to be improved in both the introduction and the discussion.
According to the Reviewer´s considerations, we have included the new English version of the manuscript and we apologize for the mistake.

Reviewer 3 Report
Interesting manuscript. It is a pity that the authors did research on one age group. Each age group is very different from each other. However, I believe that it can be assumed that in older age groups the behavior of the population is similar. It would be worth trying to study children. I believe that the work should be published. The problem of diet has long fascinated scientists. The time of COVID-19 showed all the worst habits that humanity could have at that time in the sphere of diet, movement and intellectually. Unfortunately, the lack of exercise and her diet had a huge impact on society as well as research results. enough The tables are clear and legible. References selected correctly
Author Response
Dear Reviewer, first, we want to thank you for all your contributions to our manuscript.

Round 2
Reviewer 1 Report
with these new changes the article can be published